# Generation of Bioactive Peptides from *Porphyridium* sp. and Assessment of Their Potential for Use in the Prevention of Hypertension, Inflammation and Pain

**DOI:** 10.3390/md21080422

**Published:** 2023-07-25

**Authors:** Maria Hayes, Rotimi E. Aluko, Elena Aurino, Leticia Mora

**Affiliations:** 1Department of Food BioSciences, Teagasc Food Research Centre, Ashtown, D15 DY05 Dublin, Ireland; elena.aurino@teagasc.ie; 2Department of Food and Human Nutritional Sciences, University of Manitoba, Winnipeg, MB R3T 2N2, Canada; rotimi.aluko@umanitoba.ca; 3Richardson Centre for Food Technology and Research, University of Manitoba, Winnipeg, MB R3T 2N2, Canada; 4Department of Chemical, Materials and Industrial Production Engineering, University of Naples Federico II, 80126 Naples, Italy; 5Instituto de Agroquímica y Tecnología de Alimentos (CSIC), Avenue Agustín Escardino 7, Valencia, 46980 Paterna, Spain; lemoso@iata.csic.es

**Keywords:** *Porphyridium* sp., peptides, cyclooxygenase enzymes, hypertension, inflammation, animal health, spontaneously hypertensive rats, pain relief

## Abstract

Inflammation, hypertension, and negative heart health outcomes including cardiovascular disease are closely linked but the mechanisms by which inflammation can cause high blood pressure are not yet fully elucidated. Cyclooxygenase (COX) enzymes play a role in pain, inflammation, and hypertension development, and inhibition of these enzymes is currently of great interest to researchers and pharmaceutical companies. Non-steroidal anti-inflammatory drugs are the drug of choice in terms of COX inhibition but can have negative side effects for consumers. Functional food ingredients containing cyclooxygenase inhibitors offer a strategy to inhibit cyclooxygenases without negative side effects. Several COX inhibitors have been discovered, to date, from marine and other resources. We describe here, for the first time, the generation and characterization of a bioactive hydrolysate generated using Viscozyme^®^ and Alcalase from the red microalga *Porphyridium* sp. The hydrolysate demonstrates in vitro COX-1 inhibitory activity and antihypertensive activity in vivo, assessed using spontaneously hypertensive rats (SHRs). Peptides were identified and sequenced using MS and assessed using an in silico computational approach for potential bioactivities. The peptides predicted to be bioactive, including GVDYVRFF, AIPAAPAAPAGPKLY, and LIHADPPGVGL were chemically synthesized and cyclooxygenase inhibition was confirmed. Peptides AIPAAPAAPAGPKLY and LIHADPPGVGL had COX-1 IC_50_ values of 0.2349 mg/mL (0.16 µM) and 0.2193 mg/mL (0.2 µM), respectively. The hydrolysate was included in a food carrier (jelly candies) and an antihypertensive effect was observed in SHRs.

## 1. Introduction

High blood pressure or hypertension and diseases associated with inflammation, including rheumatoid arthritis, asthma, mental illness such as depression, and gastrointestinal issues such as inflammatory bowel disease, are common conditions that may co-exist together, especially in elderly populations [1]. Inflammatory pain results in the mobilization of white blood cells and antibodies, leading to swelling and fluid accumulation. Inflammatory mediators sensitize noxious signals, pain fibers become active, and pain persists. Non-steroidal anti-inflammatory drugs (NSAIDs) such as aspirin, ibuprofen, and valdecoxib inhibit cyclooxygenase enzymes (COX-1 and COX-2 enzymes E.C.1.14.99.1) and find use in preventing inflammation and pain [2].

However, treatment with NSAIDS may have severe side effects, including ulcers, heartburn, diarrhea, and constipation. Gastrointestinal erosions and renal and hepatic insufficiency are mostly associated with COX-1 inhibitors [3]. Moreover, COX-2 inhibitors can increase blood pressure and increase the risk of atherothrombosis [4]. Natural cyclooxygenase inhibitors may provide pain relief without negative health effects. There is a need, therefore, to find new compounds to treat pain, inflammation, and hypertension without negative side effects [5].

The potential health benefits of algal oils, polysaccharides, and endogenous peptides derived from the red microalgae *Porphyridium* sp. are well known and several patents regarding their potential application as functional supplement ingredients, medicines, or topical applications exist in the literature [6,7,8]. *Porphyridium* species produce sulfated exopolysaccharides (EPSs) in a layer that surrounds the cytoplasmic membrane of the microalga and this mucilage is associated with antimicrobial, antiviral, antioxidant, and anti-inflammatory activities [9]. The protein and peptides derived from this alga also have interesting health beneficial activities. *Porphyridium* sp. are rich in proteins and make ideal candidates for generation of bioactive hydrolysates. Bioactive hydrolysates impart health benefits to the consumer that go beyond basic human nutrition based on peptides found within the hydrolysates and permeate fractions that can inhibit enzymes that play a role in human health. Peptides previously identified from protein hydrolysates can inhibit enzymes including Angiotensin-Converting Enzyme 1 (ACE-1), Renin, Dipeptidyl peptidase IV (DPP-IV), and cyclooxygenases, which play a part in the development of high blood pressure, type-2-diabetes (T2D), inflammation, and pain, respectively. Previously, the two dipeptides with amino acid single letter sequences HE and GP demonstrated anti-atherosclerotic effects on foam cell formation in the RAW264.7 cell line [10]. Peptide inhibitors of the enzyme ACE-1 are also important for the development of new alternatives to Captopril^®^ for control of hypertension, and significant research has been carried out recently concerning the development of ACE-1 inhibitory hydrolysates from dairy, microalgae, meat, seaweed, and cereal proteins [11,12,13,14,15,16].

The aim of this work is to produce a bioactive protein hydrolysate from *Porphyridium* sp. using enzymatic hydrolysis and 3 kDa molecular weight cut off (MWCO) filtration. The enzymes Viscozyme^®^ and Alcalase^®^ were used to generate hydrolysates. An in vitro bioassay found that the 3 kDa MWCO permeate from the *Porphyridium* sp. hydrolysate significantly inhibited the COX-1 enzyme compared to the positive control Resveratrol^®^. Peptides were subsequently identified from the permeate fraction using mass spectrometry. In silico analysis of identified peptides in combination with chemical synthesis, and in vitro COX-1 and COX-2 inhibition assays, identified several bioactive peptides within the hydrolysate. Furthermore, an in vivo trial performed in spontaneously hypertensive rats (SHRs) using the *Porphyridium* sp. hydrolysate alone and the hydrolysate included in the carrier food jelly sweets determined antihypertensive effects of the hydrolysate and survival of this antihypertensive activity once formulated in a food carrier. Jellies and gummies are a popular class of confectionary and product launches making health, functionality, fortified, and sustainability claims have increased [17]. In vivo results were compared to the commercially available antihypertensive drug Captopril© (positive control) and a saline solution (negative control). Results demonstrate the antihypertensive and COX-1 inhibitory bioactivities associated with a *Porphyridium* sp. hydrolysate produced in a controlled manner. Bioactive peptides with COX-1 inhibitory activities (in vitro) were identified and their enzyme inhibitory IC_50_ values are comparable to previously reported COX-1 inhibitors derived from natural products. The developed jelly sweet products have potential for use as antihypertensive products to control blood pressure and may help to ameliorate pain and inflammation in consumers. However, further in vivo trials are required to validate results obtained in vitro. 

## 2. Results

### 2.1. Proximate Compositional Analysis of Porphyridium *sp.* and Generated Hydrolysates

The protein, fat, moisture, and ash content of the whole *Porphyridium* sp. biomass was determined using AOAC methods as described in the Materials and Methods section. The protein content was 13.99% (dry weight basis), fat content was 4.82%, moisture was 9.68%, and ash content was 28.55%. A yield of 8.99% (dry weight) was obtained for the hydrolysate, which had a protein, fat, moisture, and ash content of 11.90%, 1.32%, 9.46%, and 28.96%, respectively. 

### 2.2. In Vitro Screening for Cyclooxygenase 1 and 2 Inhibition

The *Porphyridium* sp. hydrolysate inhibited COX-1 by 92.14% (±3.16) when assayed at a concentration of 1 mg/mL (Figure 1). The microalgal hydrolysate inhibited COX-2 by 32.25% when assayed at a concentration of 1 mg/mL. All assays were carried out in triplicate (*n =* 9).

Three peptides found in the *Porphyridium* sp. hydrolysate and identified using MS to have amino acid sequences GVDYVRFF, AIPAAPAAPAGPKLY, and LIHADPPGVGL with PeptideRanker scores of 0.857, 0.768, and 0.743, respectively, were assessed for their ability to inhibit COX-1 and COX-2 in vitro. The peptide GVDYDRFF inhibited COX-1 by 29.72% only. Peptides AIPAAPAAPAGPKLY and LIHAAPPGVG had COX-1 IC_50_ values of 0.2349 mg/mL (0.16 µM) and 0.2193 mg/mL (0.2 µM), respectively, based on assessment of five concentrations (1 mg/mL, 0.5 mg/mL, 0.25 mg/mL, 0.125 mg/mL, 0.0625 mg/mL) used to determine the standard curve and COX IC_50_ values (*n =* 3). However, none of the peptides inhibited COX-2 by greater than 30% when assayed at a concentration of 1 mg/mL.

### 2.3. Mass Spectrometry and In Silico Analysis of Peptides

#### 2.3.1. Peptides Identified Using Mass Spectrometry

Following MS analysis, seventy-two peptides were identified in the *Porphyridium* sp. hydrolysate generated using Viscozyme^®^ and Alcalase^®^. Only three peptide sequences were subsequently synthesized based on in silico analysis. These peptides were GVDYVRFF, corresponding to a peptide fragment found in the protein Phycoerythrin beta subunit (Accession number: tr|A0A5J4Z524|A0A5J4Z524_PORPP). The second peptide, with the amino acid sequence AIPAAPAAPAGPKLY, was derived from a putative transport protein found in *Porphyridium purpurem* (Accession number: tr|A0A5J4YZ45|A0A5J4YZ45_PORPP), and the third peptide LIHADPPGVGL corresponds with the amino acid sequence found in an uncharacterized protein found in *Porphyridium purpurem* (Accession number: tr|A0A5J4Z524|A0A5J4Z524_PORPP). These peptides were selected for synthesis based on their PeptideRanker scores but also based on the presence of amino acids such as Y, F, and L at the terminal end of the peptide.

#### 2.3.2. Determination of the PeptideRanker Scores, Novelty, and Potential Bioactivities of Peptides

PeptideRanker (http://distilldeep.ucd.ie/PeptideRanker/, accessed on 7 January 2023) is an open source software resource, used to predict the potential bioactivity of peptides based on a novel N-to-1 neural network [18]. PeptideRanker ranks the probability that a peptide sequence will be bioactive. The peptides with amino acid sequences GVDYVRFF, AIPAAPAAPAGPKLY, and LIHADPPGVGL have PeptideRanker scores of 0.857, 0.768, and 0.743, respectively (the highest scores of all identified peptides). PeptideRanker scores indicate that these peptides are likely to have bioactivity. Acceptable probability values for bioactivity are between 0.5 and 1.0. PeptideRanker scores obtained for other peptides identified in this study are shown in Table 1. Values greater than 0.5 are reported.

A search of the BIOPEP-UWM database (https://biochemia.uwm.edu.pl/biopep-uwm/, accessed on 7 January 2023) [19], UniProt Peptide Search (WoS—https://www.uniprot.org/tool-dashboard, accessed on 21 July 2023), and PubMed-indexed papers determined the novelty of the seven peptides identified and shown in Table 1. Of the peptides analyzed and listed in Table 1, their amino acid sequences were not identified in previously published papers concerning seaweed proteins and bioactive peptides, and the peptides are novel to the best of our knowledge.

A simulated digestion of identified novel peptides was performed using Peptide Cutter software (http://web.expasy.org/peptide_cutter/, accessed on 7 January 2023) [20]. This was carried out to assess if bioactive peptides could potentially survive gastrointestinal (GI) digestion or whether they are pro-peptides. Peptides shown in Table 1 were digested using enzymes found in the GI tract, including pepsin (pH 1.3), trypsin, and chymotrypsin. All peptides were cleaved into shorter peptide fragments and, in some instances, digested peptides were identified as having additional bioactivities (Table 1). Simulated GI digestion of the seven peptides produced smaller peptides such as the active dipeptide VR derived from GVDYVRFF. VR has shown ACE-1 and DPP-IV inhibitory activities [19,21,22]. Simulated digestion of the peptide LIHADPPGVGL produced the previously identified DPP-IV and DPP-III inhibitory dipeptide IH [23,24], and the ACE-1 inhibitory tri-peptide GVG was previously identified in the antihypertensive peptide VAPGVG [25].

#### 2.3.3. Chemical Synthesis of Selected Peptides

Peptides identified as having potential bioactivities with the amino acid sequences GVDYVRFF, AIPAAPAAPAGPKLY, and LIHADPPGVGL were chemically synthesized by GenScript Biotech (Leiden, The Netherlands). GenScript also verified the purity of the peptide by analytical RP-HPLC–MS. The primary structure and the theoretical values of the selected peptides were determined using PepDraw (https://www2.tulane.edu/~biochem/WW/PepDraw/, accessed on 16 June 2023) as shown in Figure 2.

### 2.4. Product Development–Production of Jelly Candies

For the development of the jelly candies containing the *Porphyridium* sp. hydrolysate, several tests were performed to determine effective combinations of different proportions of the hydrolysate, gelatin, and flavor ingredients until an appealing visual texture resulted.

### 2.5. In Vivo Antihypertensive Trial

The antihypertensive effect of the *Porphyridium* sp. hydrolysate and the jellies containing the *Porphyridium* sp. hydrolysate was evaluated. Jellies contained 3.36% (*w*/*w*) of the *Porphyridium* sp. hydrolysate. The antihypertensive effect is based on changes in hypertension physiological parameters including systolic blood pressure (SBP), diastolic blood pressure (DBP), mean arterial pressure (MAP), and heart rate (HR) after oral administration of SHRs. The *Porphyridium* sp. hydrolysate and jellies formulated with this hydrolysate reduced SBP by −1.54 mm Hg and −6.17 mm Hg, respectively, compared to Captopril^®^, which reduced SBP by −18.21 mmHg after 24 hrs. The maximal reduction of SBP observed for *Porphyridium* sp. hydrolysate was −11.67 mm Hg after 4 h and −18.97 mm Hg for *Porphyridium* sp. hydrolysate containing jellies after 2 h post oral gavage. Saline solution had no SBP-reducing effect when assessed (Figure 3a).

The *Porphyridium* sp. hydrolysate and jellies formulated with this hydrolysate reduced DBP by −2.99 mm Hg and −7.99 mm Hg, respectively, compared to Captopril^®^, which reduced DBP by −17.66 mmHg after 24 h. The maximal reduction of DBP observed for *Porphyridium* sp. hydrolysate was −7.46 mm Hg after 4 h and −15.43 mm Hg for *Porphyridium* sp. hydrolysate containing jellies after 2 h post oral gavage, compared to −27.30 mmHg for Captopril©. Saline solution again had no DBP-reducing effect when assessed (Figure 3b).

MAP was reduced by −2.31 mmHg and −7.54 mm Hg after 24 h following oral gavage of the *Porphyridium* sp. hydrolysate and jellies, respectively. This was compared to a MAP reduction of −33.7 mmHg observed following oral gavage of SHRs with Captopril© (Figure 3c). 

The decrease in HR also happened with a maximum reduction of 31 beats per min after 4 h following oral gavage of the *Porphyridium* sp. hydrolysate and a maximum reduction of 6 beats per min after 4 h following oral gavage of the *Porphyridium* sp. containing jellies and 45 beats per min after 4 h following oral gavage of Captopril©.

### 2.6. Statistical Analysis

All the experiments were performed in triplicate, and results are expressed as the mean ± standard deviation (SD) of the two replicates using EXCEL 2010. All measurements were carried out either in triplicate (*n* = 3) or (*n =* 9). Statistical analysis was performed using Excel 2010. one-way ANOVA and a post hoc Tukey’s HSD test was applied. In all cases, the criterion for statistical significance was *p* < 0.05.

## 3. Discussion

Protein was extracted from *Porphyridium* sp. biomass using a combination of Viscozyme and Alcalase. This combination of enzymes was used previously to extract proteins from seaweed biomass including *Chondrus crispus* and, indeed, microalgae such as *Nannochloropsis gaditana* [26,27]. A yield of 11.90% protein (dry weight basis) resulted. Previous studies reported protein yields using Alcalase alone of 35.1% [28]. The enzymes Alcalase, Thermolysin, and Bromelain are the most commonly used enzymes to generate bioactive peptides from biomass including algae [28]. We employed the enzyme mixture Viscozyme in our work as the mixture contains enzymes with the potential to break down the cell wall polysaccharides composed of galactose, glucose, xylose, and glucuronic acid found in *Porphyridium* species. Viscozyme contains cellulases, hemicellulases, including xylanase, and endoglucanases. Alcalase was then applied to the mixture to generate bioactive peptides from the released protein. Alcalase is widely used for this purpose in the processing and manufacture of hydrolysates containing bioactive peptides from high-protein-containing foods [29] and was applied previously to algae for this purpose by our group [30,31] and by other researchers [32].

Bioactive peptides are less than 10 kDa in size. The generated hydrolysate was passed through a 10 kDa molecular weight cut off (MWCO) filter to enrich for a permeate rich in peptides. Following MS analysis, seventy-two peptides were identified in the permeate fraction and three peptides were selected for chemical synthesis based on their novelty and predicted bioactivities using PeptideRanker. The peptides selected for synthesis, namely GVDYVRFF, AIPAAPAAPAGALLY, and LIHADPPGVGL, were not found in the BIOPEP-UWM™ database of bioactive peptides (https://biochemia.uwm.edu.pl/biopep-uwm/, accessed on 8 January 2023) or in the PepBank database (http://pepbank.mgh.harvard.edu, accessed on 8 January 2023). Analysis of these peptide sequences using PeptideRanker (http://bioware.ucd.ie/~compass/biowareweb/, accessed on 25 July 2022) and PreAIP (accessed on 19 July 2023) found that they were very likely to be bioactive as they all had scores greater than 0.74 (PeptideRanker) and many peptides were identified as having potential to be anti-inflammatory by the PreAIP program. The AF scores obtained in PreAIP for the peptides compared favorably to the score obtained for the anti-inflammatory rice peptides identified previously by Qu [33]. PeptideRanker was trained at a threshold of 0.5, i.e., any peptide predicted to have a score greater than the 0.5 threshold is labeled as bioactive. The chosen peptides were also selected based on the presence of the amino acids phenylalanine (F), tyrosine (Y), and leucine (L). The side chains of phenylalanine and leucine are long carbon chains or rings, which makes them bulky. Branched chain amino acids are essential for the cellular signaling function and when found in peptides they may be ideal for disease prevention [34]. The selected peptides have a bulky side chain aromatic amino acid at the C-terminal end of the peptide. In previously identified ACE-1 inhibitor peptides, tyrosine, phenylalanine, and tryptophan residues are often present at the C-terminus, particularly for the di-and tripeptide inhibitors [35,36]. Following simulated digestion using the Expasy peptide cutter, the peptide GVDYVRFF generated the dipeptide VR and the free amino acid F at the C-terminal end. VR was previously identified as an ACE-1 and DPP-IV inhibitory peptide [21,22]. The free amino acid phenylalanine (F) is found at the C-terminal end of previously identified ACE-I inhibitory peptides VVF, DF, and PDLVF [37,38]. AIPAAPAAPAGALLY was digested into the peptides AIPAAPAA and PAGPK and the free amino acids L and Y. GPK, found at the C-terminal end of PAGPK, occurs in eight known bioactive peptides listed in BIOPEP^TM^ and has known antibacterial and antioxidant and calcium-binding properties [39,40]. The peptide LIHADPPGVGL resulted in peptides IH and ADPPGVP and the free amino acid L following simulated digestion. The dipeptide IH is a known dipeptidyl peptidase IV (DPP-IV) inhibitory peptide [23], which indicates that the hydrolysate may also have anti-type 2 diabetes (T2D) properties following ingestion by animals. 

The rules concerning COX enzyme inhibition by bioactive peptides are not well deciphered to date. Several researchers have an interest in peptide inhibition of COX enzymes as existing COX inhibitors, such as the non-steroidal anti-inflammatory drugs (NSAIDs) rofecoxib and valdecoxib, which are known cardiovascular toxins and so have been withdrawn from the market [41]. Over 140 dipeptides were previously screened for their ability to inhibit COX-2 [42]. In this work, the *Porphyridium* sp. hydrolysate inhibited COX-1 by 92.14% (±3.16) and the chemically synthesized peptides AIPAAPAAPAGPKLY and LIHADPPGVGL had COX-1 IC_50_ values of 0.2349 mg/mL (0.16 µM) and 0.2193 mg/mL (0.2 µM), respectively, when assayed at 1 mg/mL concentrations compared to the positive control Resveratrol^®^. The peptide GVDYVRFF did not inhibit COX-1 significantly. COX-2 was not significantly inhibited by the *Porphyridium* sp. hydrolysate or the synthesized peptides identified in the hydrolysate. Natural-product-based COX 1 inhibitors discovered to date seem better than NSAIDs in terms of side effects such as thrombotic cardiovascular events (COX-2 inhibitors) or gastrointestinal erosions, and renal and hepatic insufficiency (COX-1 inhibitors). Natural compound inhibitors of COX-1 and COX-2 are generally supposed to be devoid of severe side effects. The COX-1 IC_50_ values obtained for AIPAAPAAPAGPKLY and LIHADPPGVGL compare favorably with COX IC_50_ values previously determined for natural product COX-1 inhibitors. Previously, researchers identified a peptide named RQ-9 with the sequence RLARAGLAQ that was found to inhibit COX activity with a COX-1 IC_50_ value of 0.31 µg/mL and a COX-2 IC_50_ value of 4.77 µg/mL [43].

COX-1 inhibitors are associated with side effects including heart health issues. In this study, a *Porphyridium* sp. hydrolysate with COX-1 inhibitory activity was fed to SH rats and no adverse effects were observed. Antihypertensive activity resulted. The SH rat is a well-established model of human essential hypertension. Various studies to determine the antihypertensive effects of food-derived bioactive peptides have used spontaneous hypertensive animals as a model system [44,45]. A decrease in SBP and DBP as well as MAP resulted, following consumption of the hydrolysate and hydrolysate included at 3.36% (total weight) in jelly candies, up to 6 h post consumption and compared to the positive control Captopril©. This result compares favorably with results obtained concerning antihypertensive activity in SH rats observed following administration by the oral route of hydrolysates of royal jelly where blood pressure slowly decreased with time by about 15 mmHg from the previous level after 8 h in the control group [46]. The food carrier used in this study (jelly candies) proved effective at maintaining the bioactivity of the *Porphyridium* sp. hydrolysate and actually increased antihypertensive activity by ~20% compared to the hydrolysate alone in relation to SBP and DBP. This may be due to release of gelatin/collagen peptides following digestion in the SHR that could further enhance antihypertensive activity. However, we did not include hydrolyzed gelatin as a control. A study carried out recently by Cao and colleagues [47] identified that bovine-gelatin-derived peptides (BGPs) reduced blood pressure, triglyceride levels, and the low-density lipoprotein cholesterol/high-density lipoprotein cholesterol ratio in SHRs. This was achieved through downregulation of angiotensin converting enzyme (ACE-1), angiotensin II (Ang II), and Ang II type 1 receptor (AT1R) levels and upregulation of Ang II type 2 receptor (AT2R) levels. *Porphyridium* sp. were previously shown to lower plasma cholesterol levels in animals through different physiological mechanisms [33].

## 4. Materials and Methods

### 4.1. Porphyridium *sp.* Hydrolysate and Permeate Generation

Freeze-dried *Porphyridium* sp. biomass was cultivated and processed by Thomas More (Geel, Belgium) and VITO (Mol, Belgium). To freeze dry, *Porphyridium* sp. microalgae were suspended in ddH_2_O (10%, *w*/*v*) and placed in a water bath (Grant JB Aqua 12 water bath, Grant instrument, England, UK) at 80 °C for 10 min to deactivate endogenous enzymes present in the biomass. The pH of the mixture was adjusted using 1M HCl and the enzyme Viscozyme was added to the sample (1% *w*/*w*). The mixture was incubated at 45 °C for 2 h with stirring at 220 rpm in a shaking incubator. Samples were removed from the incubator after 2 h and subsequently were heat-deactivated in a water bath at 80 °C × 10 min. The pH of the mixture was adjusted to 8–8.5 using 1M NaOH and the enzyme Alcalase was added to the mixture. The temperature was maintained at 55 °C and stirred at 220 rpm. To assess the efficiency of the enzymatic process, the permeate yield, protein recovery, and the degree of hydrolysis (DH) were calculated as follows:$$\%\;permeate\;yield\;[g/g]\;=\frac{mass\;of\;permeate}{mass\;of\;whole\;biomass}\times 100$$
$$\%\;protein\;recovery\;[g/g]\;=\frac{mass\;of\;protein\;in\;the\;permeate}{mass\;of\;protein\;in\;the\;whole\;biomass}\times 100$$

The degree of hydrolysis (DH) was calculated at the end of each hydrolysis stage using the trichloroacetic acid (TCA) method described previously by Hoyle and colleagues [48]. A sample of 1 mL of the hydrolysate was collected after the deactivation step and added to 1 mL of 20% (*w*/*v*) TCA. The solutions were left to settle for 30 min and then centrifuged at 7800× *g* × 15 min. The percentage protein in the supernatant and the hydrolysate sample were assessed using the QuantiPro BCA Assay kit (Sigma, St. Louis, MO, USA) as per the manufacturer’s instructions and DH was calculated as follows:$$DH\%=\frac{TCAsoluble\;N\;in\;the\;supernatant}{total\;N\;in\;the\;hydrolysate}\times 100$$

The hydrolysate obtained was filtered using a 3 kDa molecular weight cut-off (MWCO) membrane filter (Millipore, Tullagreen, Carrigtwohill, Co. Cork, Ireland), and a permeate and a retentate fraction recovered.

Proximate compositional analysis was performed on the whole biomass and hydrolysates. The nitrogen percentage in the samples was determined using the LECO FP628 (LECO Corp., St Joseph, MI, USA) protein analyzer by applying the Dumas AOAC method 992.15 (1990) [49]. The protein content was obtained using a conversion factor of 6.25. The percentage lipid in each sample was assessed using the Oracle NMR Smart Trac rapid Fat analyzer (CEM Corporation, Matthews, NC, USA) using AOAC official methods 985.14. The ash and moisture content of the samples was determined as described previously [49].

### 4.2. In Vitro Bioactivities Assessment

#### 4.2.1. In Vitro Screening for Cyclooxygenase (COX-1 and COX-2) Inhibitory Activities

In vitro assays to determine COX inhibition were performed using cell-free assays as described previously [30,50]. The hydrolysate or peptides were incubated independently with ovine recombinant COX-1 (Cayman Chemicals, Hamburg, Germany) or human recombinant COX-2 (Cayman Chemicals, Hamburg, Germany). Resveratrol (Merck, Dublin, Ireland) was used as a positive control. The assay was performed in accordance with the manufacturers’ instructions. Briefly, the assay measures the peroxidase components of COXs in the presence and absence of COX inhibitors or test components. Peroxidase activity is measured colorimetrically and the appearance of oxidized N, N, N’, N’-tetramethyl-p-phenylenediamine (TMPD) is observed at 590 nm. The percentage of COX-1 or COX-2 inhibition is calculated based on the increase or decrease in absorbance observed at 590 nm in the presence or absence of the test component (control or test). The assay is carried out using a 96-well plate and includes a background well containing 160 µL of the assay buffer (0.1 M Tris-HCL, pH8) and 10 µL of Hemin. The 100% initial activity is observed in triplicate and wells contain 150 µL of the assay buffer, 10 µL of Hemin, and 10 µL of either COX-1 or COX-2 enzymes. Inhibitor wells contain 150 µL of assay buffer, 10 µL of Hemin, and 10 µL of enzyme, and 10 µL of the test inhibitor (algal extract) or the positive control (Resveratrol). Test compounds were dissolved in dimethyl sulfoxide (DMSO). Following addition of all elements to the wells, the plate is incubated at 25 °C for two minutes. A colorimetric substrate is added to all wells and immediately 20 µL of arachidonic acid is subsequently added. The absorbance of all wells is measured at 590 nm.

The average absorbance of all wells is calculated. The absorbance of the background wells is subtracted from all wells. The absorbance value for the test sample is subtracted from the initial activity absorbance value and subsequently this is divided by the 100% initial activity absorbance value. The percentage COX-1/COX-2 inhibition is calculated by multiplying this final value by 100. COX-1 and COX-2 IC_50_ values relate to the concentrations that give 50% inhibition. 

#### 4.2.2. Mass Spectrometry in Tandem and In Silico Analysis

The *Porphyridium* sp. 3 kDa MWCO permeate fraction was processed for mass spectrometry analysis using the Preomics Phoenix Clean-up Kit (96×), (Preomics, D-82152 Planegg/Martinsried, Germany) in accordance with the manufacturers’ instructions and as described previously [30]. The sample was acidified and hydrophobic and hydrophilic contaminants removed using a series of wash steps and peptides eluted from the cartridge and prepared in loading buffer for LC-MS/MS analysis. Peptides were identified using a mass spectrometer nanoESI qQTOF (6600 plus TripleTOF, AB SCIEX, Framingham, MA, USA) using liquid chromatography and tandem mass spectrometry (LC–MS/MS). A total of 1 μL of microalgal permeate was loaded onto a trap column (3 µ C18-CL 120 Ᾰ, 350 μM × 0.5 mm; Eksigent, Redwood City, CA, USA) and desalted with 0.1% TFA (trifluoroacetic acid) at 5 µL/min for 5 min. The peptides were then loaded onto an analytical column (3 µ C18-CL 120 Ᾰ, 0.075 × 150 mm) equilibrated in 5% acetonitrile (ACN) 0.1% formic acid (FA). Elution was carried out with a linear gradient from 5 to 40% B in A for 20 min, where solvent A was 0.1% FA and solvent B was ACN with 0.1% FA) at a flow rate of 300 nL/min. The sample was ionized in an electrospray source of Optiflow < 1 μL Nano applying 3.0 kV to the spray emitter at 200 °C. Analysis was carried out in a data-dependent mode. Survey MS1 scans were acquired from 350 to 1400 *m*/*z* for 250 ms. The quadrupole resolution was set to ‘LOW’ for MS2 experiments, which were acquired from 100 to 1500 *m*/*z* for 25 ms in ‘high sensitivity’ mode. The following switch criteria were used: charge: 1+ to 4+; minimum intensity; 100 counts per second (cps). Up to 50 ions were selected for fragmentation after each survey scan. Dynamic exclusion was set to 15 s. The system sensitivity was controlled by analyzing 500 ng of K562 protein extract digest (SCIEX); in these conditions, 1819 proteins were identified (FDR < 1%) in a 20 min gradient. Protein Pilot v 5.0. (SCIEX Framingham, MA, USA) default parameters were used to generate the peak list directly from 6600 plus TripleTOF wiff files. The Paragon algorithm of ProteinPilot v 5.0 was used to search the SwisProt and Uniprot Prot-Algae database with the following parameters: no enzyme specificity, IAM cys-alkylation, no taxonomy restriction, and the search effort set to throughout. Peptides were identified with a confidence of ≥95%.

The potential of identified peptides to be bioactive was predicted using the PeptideRanker tool [18] (http://bioware.ucd.ie/~compass/biowareweb/, accessed on the 7 January 2023) and peptide scores were obtained. The novelty of identified peptides was determined by performing a literature and database search in BIOPEP-UWM (http://www.uwm.edu.pl/biochemia/index.php/en/biopep, accessed on 3 January 2023). In addition, peptide sequences were assessed using the anti-inflammatory peptide predictor PreAIP (http://kurata1.bio.kyutech.ac.jp/PreAIP/, accessed on 17 July 2023) [51].

### 4.3. Chemical Synthesis of Peptides

Bioactive peptides predicted to be bioactive using PeptideRanker and with the amino acid sequences GVDYVRFF, AIPAAPAAPAGPKLY, and LIHADPPGVGL were chemically synthesized by GenScript Biotech (Leiden, The Netherlands). GenScript also verified the purity of the peptide by analytical RP-HPLC–MS. The primary structure and the theoretical values of the selected peptides was determined using PepDraw (https://www2.tulane.edu/~biochem/WW/PepDraw/, accessed on 8 January 2023).

### 4.4. Product Formulation

Ingredients used for jelly candy production included: 0.5 g of the hydrolysate (3.36% of dry weight ingredients), 0.75 g agar (Merck, Dublin, Ireland), 0.75 g pectin (Merck, Dublin, Ireland), 0.75 g of guar gum (Sigma Aldrich, Dublin, Ireland), 0.2 mL of liquid sucrose (Dr Oetker, Dublin 12, Ireland), 0.5 mL of liquid food coloring (pink) (Dr Oetker, Dublin 12, Ireland), 0.5 mL orange extract (Dr Oetker, Dublin 12, Ireland), 3.5 sheets (approximately 12 g) of sheet bovine derived gelatin (Dr Oetker, Dublin 12, Ireland), 57 mL of ddH_2_O. The jelly candies (jellies) were manufactured as described previously [17,52]. A cold-set approach was used. Once achieved, the mixture was poured into polyethylene molds (2 × 1.5 cm) and refrigerated for 24 h at between 2 and 8 °C. The formulation used for the jelly candies was as follows: 6 g of *Porphyridium* sp. hydrolysate, 570 mL of ddH_2_O, 0.75 g agar, 0.75 g pectin, 0.75 g guar gum, 5 g liquid sucrose, 1 g ascorbic acid, 1 mL food coloring (red), 1 mL of lemon extract essence, and 12 g of bovine gelatin (~3.5 sheets). All the *Porphyridium* sp. jellies produced were removed from the molds and stored in a freezer at −80 °C until the time of further analysis. The bioactive hydrolysate was added at a concentration of 3.36% (*w*/*v*) to the formulation. Jellies were rolled in cornstarch to prevent sticking following removal from molds. 

### 4.5. In Vivo Trial in SHRs

*Animal model and experimental design*: Spontaneously hypertensive rats (SHRs), implanted with Data Sciences International (DSI) HD-S10 telemetry transmitters (DSI, St. Paul, MN, USA), were purchased from Envigo RMS, LLC (Indianapolis, IN, USA) and used without further alteration. The rats were housed under a 12 h day and night cycle at 21 °C with regular chow feed and tap water provided ad libitum. Animal experiments were carried out following the Canadian Council on Animal Care Ethics guidelines with a protocol approved by the University of Manitoba Fort Garry Campus Animal Care Committee. For each experiment, there were four rats per group: phosphate buffered saline (PBS), Captopril (50 mg/kg body weight (wt) dissolved in PBS), and sample (100 mg/kg body wt dissolved in PBS). Each rat was orally gavaged with a 1 mL solution using a disposable plastic syringe; blood pressure was then recorded continuously in freely moving rats for 24 h by telemetry. Results are reported as changes in values of the SBP, DBP, MAP, and HR at different time points over a 24 h period minus their baseline measurements at time zero.

*Telemetry recording and signal processing*: Systolic and diastolic blood pressure (SBP and DBP) measurements were performed in a quiet room with each rat cage placed on top of one receiver (Model RPC-1, DSI instruments, St. Paul, MN, USA). Real time experimental data (including heart rates (HRs)) were recorded continuously using the Ponemah 6.1 data acquisition software (DSI instruments, MN, USA). The system was attached to an APR-1 atmospheric-pressure monitor (DSI instruments, MN, USA), which normalized the transmitted pressure values and ensured that the recorded blood pressure signals were not dependent on changes in atmospheric pressure. The zero time values for each measured parameter were then subtracted from the respective values obtained at 2, 4, 6, 8, and 24 h, where the most notable changes were observed. The rat experiments were performed in four replicates and standard deviations were calculated using Excel.

## 5. Conclusions

Bioactive peptides were successfully generated from the red microalga *Porphyridium* sp. and displayed ACE-1 and COX-1 inhibitory activities in vitro. In vivo studies in SHRs confirmed an antihypertensive effect of this hydrolysate and additionally, the antihypertensive effect of the hydrolysate was enhanced when included in a gelatin carrier at a concentration of 3.36%. Additional bioactivities were assigned to peptides found in the hydrolysate using in silico methods including PreAIP and Peptide Ranker. The hydrolysate ingredient and the gelatin confectionary hold potential for use in the prevention of high blood pressure and pain in humans and animals but further in vivo trials are required to validate these findings.

## Figures and Tables

**Figure 1 marinedrugs-21-00422-f001:**
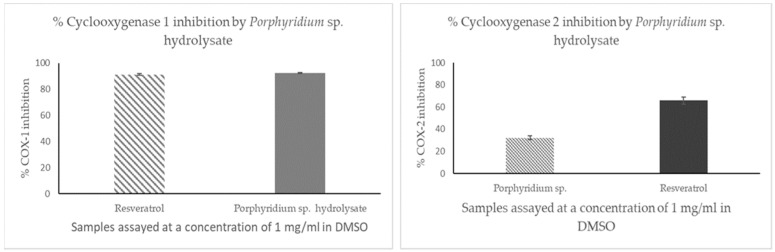
Cyclooxygenase (COX-1 and COX-2) inhibition by the *Porphyridium* sp. hydrolysate assayed at a concentration of 1 mg/mL in DMSO. Control used was Resveratrol, a known COX-1 inhibitor, assayed at 1 mg/mL concentration (*n =* 9).

**Figure 2 marinedrugs-21-00422-f002:**
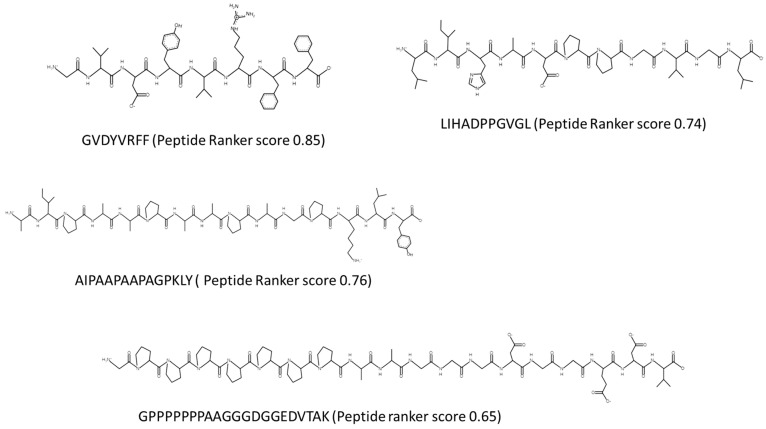
Chemical structure and characteristics of synthesized peptides derived from the *Porphyridium* sp. hydrolysate. Chemical structures were drawn using PepDraw (PepDraw (tulane.edu))–accessed on 16 June 2023.

**Figure 3 marinedrugs-21-00422-f003:**
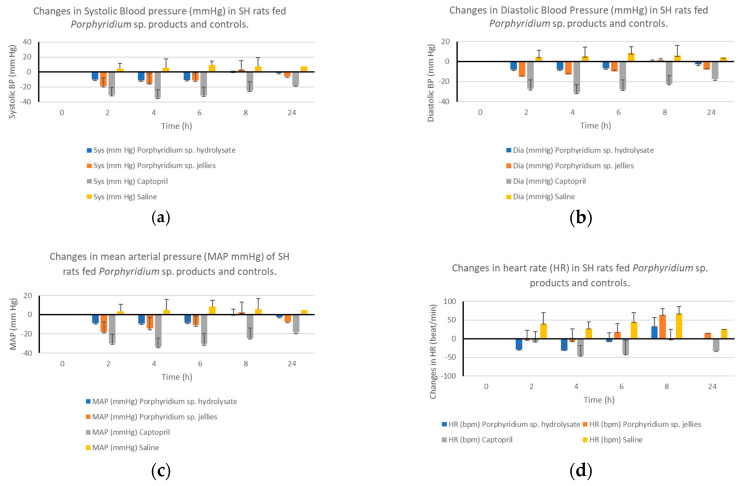
(**a**): The effect of the *Porphyridium* sp. hydrolysate and jellies containing 3.36% dry weight of hydrolysate on SBP compared to the positive control Captopril© and a saline solution (negative control). (**b**): The effect of the *Porphyridium* sp. hydrolysate and jellies containing 3.36% dry weight of hydrolysate on DBP compared to the positive control Captopril© and a saline solution (negative control). (**c**): Reductions in MAP (mm Hg) observed following oral gavage of *Porphyridium* sp. hydrolysate, jellies containing the hydrolysate, saline, and the positive control Captopril©. (**d**): Changes in HR following oral gavage of the *Porphyridium* sp. hydrolysate, jellies, and controls.

**Table 1 marinedrugs-21-00422-t001:** Peptide sequences identified from *Porphyridium* sp. hydrolysate permeate fraction generated using Viscozyme, Alcalase, and bioactivities of peptide fragments obtained after simulated gastrointestinal digestion.

*Porphyridium* sp. Peptides Generated from Hydrolysis								
Peptide Amino Acid Sequence (Single Letter Code)	Protein of Origin	Accession Code	Peptide Ranker Value *	PreAIP RF *^†^ Combined Values	BIOPEP Search **	Expasy Peptide Cutter ***	Bioactivities Associated with Peptide Fragments	Peptide Charge
GVDYVRFF	Uncharacterized protein OS = *Porphyridium purpureum* OX = 35,688 GN = FVE85_6371 PE = 4 SV = 1	tr|A0A5J4Z524|A0A5J4Z524_PORPP	0.857	0.374 Low confidence	Novel	GVD, Y, VR, F, F	GVD-alpha amylase inhibitor; VR-DPP-IV and ACE inhibitor peptide	0
AIPAAPAAPAGPKLY	Putative transporter YrhG OS = *Porphyridium purpureum* OX = 35,688 GN = FVE85_0107 PE = 4 SV = 1	tr|A0A5J4YZ45|A0A5J4YZ45_PORPP	0.768	0.262 Negative AIP	Novel	AIPAAPAAPAGPK; L, Y	All novel but GPK occurs at the C-terminal end of antioxidative, antibacterial, osteoanabolic and calcium binding peptides found in BIOPEP	1
LIHADPPGVGL	Uncharacterized protein OS = *Porphyridium purpureum* OX = 35,688 GN = FVE85_6371 PE = 4 SV = 1	tr|A0A5J4Z524|A0A5J4Z524_PORPP	0.743	0.390 Medium confidence AIP	Novel	L, IH, ADPPGVG, L	IH-DPP-IV inhibitor & DPP-III inhibitor; GVG occurs in alpha amylase and ACE-I inhibitors	−1
GLDAGLSHCGVVNVCIP	UDP-glucose/GDP-mannose dehydrogenase OS = *Ectocarpus siliculosus* OX = 2880 GN = Esi_0052_0113 PE = 4 SV = 1	tr|D8LPB4|D8LPB4_ECTSI	0.729	0.500 High confidence AIP	Novel	G, L, DAG, L, SH, CGVVNVCIP	DAG-found in ACE-1 and antioxidative peptides; SH is a DPP-IV inhibitory peptide	−1
LIHADPPGVGLTGF	Uncharacterized protein OS = *Porphyridium purpureum* OX = 35,688 GN = FVE85_6371 PE = 4 SV = 1	tr|A0A5J4Z524|A0A5J4Z524_PORPP	0.716	0.369 low confidence AIP	Novel	L, IH, ADPPGVG, L, TG, F	IH- DPP-IV inhibitor and DPP-III inhibitor, GVG occurs in alpha amylase and ACE-1 inhibitor peptides, TG-DPP-IV inhibitor and ACE-1 inhibitor	−1
AIPAAPAAPAGPK	Putative transporter YrhG OS = *Porphyridium purpureum* OX = 35,688 GN = FVE85_0107 PE = 4 SV = 1	tr|A0A5J4YZ45|A0A5J4YZ45_PORPP	0.709		Novel	AIPAAPAAPAGPK	All novel but GPK occurs at the C-terminal end of antioxidative, antibacterial, osteoanabolic and calcium binding peptides found in BIOPEP	1
GPPPPPPPAASGGDGGEDVTAK	Adenylyl cyclase-associated protein 2 OS = *Porphyridium purpureum* OX = 35,688 GN = FVE85_5192 PE = 3 SV = 1	tr|A0A5J4Z4T9|A0A5J4Z4T9_PORPP	0.659	0.469 High confidence AIP	Novel	Resistance to cleavage by Trypsin, Chymotrypsin and Pepsin	TAK found in the immunomodulator peptide RTAKV	−2
AAGGSLFEEYMR	Protein PYP1 OS = *Porphyridium purpureum* OX = 35,688 GN = FVE85_6364 PE = 4 SV = 1	tr|A0A5J4Z470|A0A5J4Z470_PORPP	0.648	0.461 Medium confidence AIP	Novel	AAGGS, L, F, EE, Y, M, R	GGS found in the sequence of an ACE-1 inhibitory peptide YAGGS	−1
LFDGKVCTMLIIIT	NADH-ubiquinone oxidoreductase chain 5 OS = *Rhodogorgon* sp. OX = 2,485,824 GN = ND5 PE = 3 SV = 1	tr|A0A3G3MIM0|A0A3G3MIM0_9FLOR	0.641	0.545 High confidence AIP	Novel	L, F, DGK, VCTM, L, IIIT	all novel peptide fragments- no assigned bioactivities but IT is found in the ACE-1 inhibitory peptides ITT and ITK	0
LDSHLPINLPQGL	Ribulose bisphosphate carboxylase large chain (Fragment) OS = *Porphyridium purpureum* OX = 35,688 GN = rbcL PE = 3 SV = 1	tr|Q1WFR3|Q1WFR3_PORPP	0.619	0.428 Medium confidence AIP	Novel	L, DSH, L, PINL, PQG, L	PQG occurs at the end of the PEP inhibitory peptide PPPPGGPQPRPPQG	−1

* PeptideRanker http://bioware.ucd.ie/~compass/biowareweb/, accessed on 25 July 2022; PreAIP ***^†^** http://kurata14.bio.kyutech.ac.jp/PreAIP/, accessed on 19 July 2023; ** BIO-PEP-UWM https://biochemia.uwm.edu.pl/biopep-uwm/, accessed on 8 January 2023; *** PeptideCutter-SIB Swiss Institute of Bioinformatics Expasy, accessed on 7 January 2023.

## Data Availability

Further data is available from the corresponding author.

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
