# Peer review of "Generation of Bioactive Peptides from Porphyridium sp. and Assessment of Their Potential for Use in the Prevention of Hypertension, Inflammation and Pain"

_marinedrugs, 2023, doi:10.3390/md21080422_

Round 1
Reviewer 1 Report
In this study, the authors reported the anti-COX and anti-hypertensive activity of a protein hydrolysate generated from the red microalga Porphyridium sp. and selected peptides identified from the hydrolysate. Strength of the study is that the authors have combined in vitro, in silico and in vivo methods in their investigation. Writing-wise, the authors should consider amending CONCLUSION (lines 473-475), which looks strange. In addition, certain parts of the RESULTS sound like M&M (e.g., lines 186-193) and Discussion (e.g., lines 236-242). Please check the rest and revise where appropriate.
Below are my other feedbacks for the authors’ consideration:
1. INTRODUCTION: Line 78: “… significantly inhibited the COX-1 enzyme relevant to the positive control Resveratrol®” – The use of the word “relevant” is confusing/unclear here.
2. RESULTS:
(a) Line 111 vs line 115: Did the authors use 3 or 9 replicates? Please check.
(b) Figure 1: Authors should be careful when describing or interpreting the results presented in Figure 1, particularly when comparing the activity of the hydrolysate to that of resveratrol. The data only originated from a single concentration level, with no EC50 data provided. So, it is unclear whether the activity data originated from a linear segment within the activity range or from a portion where the activity has reached a plateau.
(c) Figure 1: Please minimize or eliminate repetitive information found in the figure caption, x-axis label, and chart titles. Furthermore, the charts themselves are confusing due to the inconsistent ordering of resveratrol and hydrolysate bars between the two charts. Additionally, for the sample, one chart says "Polyphyridium sp. hydrolysate," while the second chart says "Polyphyridium sp."
(d) Line 122: “five concentrations” – What were they?
(e) Line 123: “However, none of the peptides inhibited COX-2 when compared to Resveratrol” – Meaning unclear. Did the authors mean the peptides showed 0% activity or their activities were lower than that of resveratrol?
(f) Lines 140-148: PeptideRanker is a rather generic tool. Since the authors were targeting anti-inflammatory peptides, why didn’t the authors use some of the freely available online tools (e.g., PreAIP) that can be used for predicting anti-inflammatory peptides? These tools would be more specific compared with PeptideRanker. Alternatively, using molecular docking to screen for anti-COX peptides may be another promising approach.
(g) Line 149: The authors should be very careful when concluding whether their sequences are novel simply based on their search in the BIOPEP-UWM database. There are some peptide sequences reported in Scopus-, WoS-, and/or PubMed-indexed papers which have not been deposited into the BIOPEP-UWM database.
(h) Figures 3a-3d: Please revise them so that the x-axis labels are not overlapping with the bars. The results also appear incomplete – did the authors run any statistical analysis on the data? If so, info about statistical significance should be indicated clearly in the charts.
(i) Lines 239-242: “Inclusion of the hydrolysate in jelly candies enhanced … further analysis of the jellies is required to confirm this.” – This statement seems more suitable for DISCUSSION than for RESULTS. In Lines 321-326 of DISCUSSION, a similar statement is provided. So, the authors should consider consolidating them into one in the DISCUSSION. On the other hand, regarding the possibility that gelatin degradation might release anti-hypertension peptides in vivo – did the authors consider treating the animal model with only the jellies (not containing protein hydrolysates)? That could provide some indication as to whether the enhanced effects exerted by jellies+protein hydrolysate can be attributed to gelatin degradation.
3. DISCUSSION:
(a) Line 247: “A satisfactory protein extract yield (11.90%) resulted” – Why is 11.90% considered “satisfactory”? Did the authors come across any info in the literature that support this statement?
(b) Lines 248-249: “Thermolysin and Bromelain are the most commonly used enzymes to generate bioactive peptides from biomass including algae” – Can the authors clarify why they did not use these two enzymes instead since they have been used commonly, even on algae?
(c) Lines 240-247: “Chosen peptides were also chosen based on the presence of the amino acids phenylalanine (F), Tyrosine (Y) and Leucine (L)…” – It is unclear how choosing such sequences contribute towards the aim of the study to search for anti-inflammatory/anti-COX/anti-hypertension/pain-reducing peptides. Please improve the rationale statement.
4. M&M: Please include a short description/section to provide information about the statistical tests that were conducted, the software used, etc.
Author Response
Reviewer comment 1: Comments and Suggestions for Authors
In this study, the authors reported the anti-COX and anti-hypertensive activity of a protein hydrolysate generated from the red microalga Porphyridium sp. and selected peptides identified from the hydrolysate. Strength of the study is that the authors have combined in vitro, in silico and in vivo methods in their investigation. Writing-wise, the authors should consider amending CONCLUSION (lines 473-475), which looks strange. In addition, certain parts of the RESULTS sound like M&M (e.g., lines 186-193) and Discussion (e.g., lines 236-242). Please check the rest and revise where appropriate.
Response: We thank the reviewer for their comments and we agree with the reviewer that the conclusion section lines 473-475 could be strengthened. In line with the request from the reviewer we have edited lines 473-475 as follows: “
We have removed lines 186-193 from the results section and rewritten the materials and methods section as follows: “Ingredients used for jelly candy production included: 0.5 g of the hydrolysate (3.36 % of dry weight ingredients), 0.75 g agar (Merck, Dublin, Ireland), 0.75 g pectin (Merck, Dublin, Ireland), 0.75 g of guar gum (Sigma Aldrich, Dublin, Ireland), 0.2 ml of liquid sucrose (Dr Oetker, Dublin 12, Ireland), 0.5 ml of liquid food coloring (pink) (Dr Oetker, Dublin 12, Ireland), 0.5 mL orange extract (Dr Oetker, Dublin 12, Ireland), 3.5 sheets (approximately 12 g) of sheet bovine derived gelatin (Dr Oetker, Dublin 12, Ireland), 57 mL of ddH2O. The jelly candies (jellies) were manufactured as described previously [17, 29]. A cold set approach was used. Once achieved, the mixture was poured into polyethylene moulds (2 x 1.5 cm) and refrigerated for 24 h at between 2-8°C. The formulation used for the jelly candies was as follows 6 g of Porphyridium sp. hydrolysate, 570 mL of ddH2O, 0.75 g agar, 0.75 g pectin, 0.75 g guar gum, 5 g liquid sucrose, 1 g ascorbic acid, 1 mL food colouring (red), and 1 mL of lemon extract essence and 12 g of bovine gelatine (~3.5 sheets). All the Porphyridium sp. jellies produced were removed from the moulds and stored in a freezer at -80 °C until the time of further analysis. The bioactive hydrolysate was added at a concentration of 3.36% (w/v) to the formulation. Jellies were rolled in cornstarch to prevent sticking following removal from molds.”
Additionally, we have removed lines 236-242.
Below are my other feedbacks for the authors’ consideration:
- INTRODUCTION: Line 78: “… significantly inhibited the COX-1 enzyme relevant to the positive control Resveratrol®” – The use of the word “relevant” is confusing/unclear here.
Response to reviewer comments: We have removed the word “relevant” from the sentence as requested by the reviewer and replaced it with compared to Resveratrol®.
- RESULTS:
(a) Line 111 vs line 115: Did the authors use 3 or 9 replicates? Please check.
Response: We thank the reviewer for picking up this error – we used n=9 and have corrected the text accordingly.
(b) Figure 1: Authors should be careful when describing or interpreting the results presented in Figure 1, particularly when comparing the activity of the hydrolysate to that of resveratrol. The data only originated from a single concentration level, with no EC50 data provided. So, it is unclear whether the activity data originated from a linear segment within the activity range or from a portion where the activity has reached a plateau.
Response: We agree with the reviewer here and have revised the text to reflect this. We carried out the assay using one concentration level. The text now reflects this – we refer to Figure 1.
It reads as follows: “The Porphyridium sp. hydrolysate inhibited COX-1 by 92.14 % (+ 3.16) when assayed at a concentration of 1 mg/mL (Figure 1). The microalgal hydrolysate inhibited COX-2 by 32.25 % when assayed at a concentration of 1 mg/ml. All assays were carried out in triplicate (n=9).”
(c) Figure 1: Please minimize or eliminate repetitive information found in the figure caption, x-axis label, and chart titles. Furthermore, the charts themselves are confusing due to the inconsistent ordering of resveratrol and hydrolysate bars between the two charts. Additionally, for the sample, one chart says "Polyphyridium sp. hydrolysate," while the second chart says "Polyphyridium sp."
Response: We always label our figures in this manner with a clear x and y axis label, a chart heading label. We would prefer to leave this figure as is.
(d) Line 122: “five concentrations” – What were they?
Response: We have highlighted the five concentrations used in the following text:
“Peptide GVDYDRFF inhibited COX-1 by 29.72% only. Peptides AIPAAPAAPAGPKLY and LIHAAPPGVG had COX-1 IC50 values of 0.2349 mg/ml (0.16 µM) and 0.2193 mg/ml (0.2 µM) respectively based on assessment of five concentrations (1 mg/ml, 0.5 mg/ml, 0.25 mg/ml, 0.125 mg/ml, 0.0625 mg/ml) used to determine the standard curve and COX IC50 values (n=3). However, none of the peptides inhibited COX-2 by greater than 30 % when assayed at a concentration of 1 mg/ml.”
(e) Line 123: “However, none of the peptides inhibited COX-2 when compared to Resveratrol” – Meaning unclear. Did the authors mean the peptides showed 0% activity or their activities were lower than that of resveratrol?
Response: We agree with reviewer 1 that this sentence is confusing and we have rewritten it in the text to provide clarity. The sentence now reads as follows: “However, none of the peptides inhibited COX-2 by greater than 30 % when assayed at a concentration of 1 mg/ml.”
(f) Lines 140-148: PeptideRanker is a rather generic tool. Since the authors were targeting anti-inflammatory peptides, why didn’t the authors use some of the freely available online tools (e.g., PreAIP) that can be used for predicting anti-inflammatory peptides? These tools would be more specific compared with PeptideRanker. Alternatively, using molecular docking to screen for anti-COX peptides may be another promising approach.
Response: We actually did not know about PreAIP and agree completely with the reviewer and thank them for bringing this to our knowledge. In the revised paper, we have now included the PreAIP predictions of anti-inflammatory activity related to the identified peptides. We have edited the table to reflect the search and we have edited the text additionally. We also added reference [52] - Khatun, M. S., Hasan, M. M., Kurata, H. (2019). PreAIP: Computational prediction of anti-inflammatory peptides by integrating multiple complementary features. Frontiers in Genetics, 10, 129, doi: https://10.3389/fgene.2019.00129
(g) Line 149: The authors should be very careful when concluding whether their sequences are novel simply based on their search in the BIOPEP-UWM database. There are some peptide sequences reported in Scopus-, WoS-, and/or PubMed-indexed papers which have not been deposited into the BIOPEP-UWM database.
Response: We agree with reviewer 1 on this and in the text we highlighted that we did an additional search of the literature to see if identified peptides were novel or not (to the best of our knowledge). In the revised manuscript, we have edited the text to include a search in WoS and PubMed indexed papers. The text reads as follows now:
“A search of the BIOPEP-UWM database (https://biochemia.uwm.edu.pl/biopep-uwm/, accessed on 7th of January 2023) [19], UniProt Peptide Search (WoS – https://www.uniprot.org/tool-dashboard – accessed 21/07/2023) and PubMed-indexed papers determined the novelty of the seven peptides identified and shown in Table 1. Of the peptides analyzed and listed in Table 1, their amino acid sequences were not identified in previously published papers concerning seaweed proteins and bioactive peptides and the peptides are novel to the best of our knowledge.”
(h) Figures 3a-3d: Please revise them so that the x-axis labels are not overlapping with the bars. The results also appear incomplete – did the authors run any statistical analysis on the data? If so, info about statistical significance should be indicated clearly in the charts.
Response: We have revised the figures 3a-3d so that the x-axis label is not overlapping with the bars. In terms of statistical analysis we include the following in the revised text Lines 475-478: “The rat experiments were performed in four replicates and standard deviations calculated using Excel.
- Lines 239-242: “Inclusion of the hydrolysate in jelly candies enhanced … further analysis of the jellies is required to confirm this.” – This statement seems more suitable for DISCUSSION than for RESULTS. In Lines 321-326 of DISCUSSION, a similar statement is provided. So, the authors should consider consolidating them into one in the DISCUSSION. On the other hand, regarding the possibility that gelatin degradation might release anti-hypertension peptides in vivo – did the authors consider treating the animal model with only the jellies (not containing protein hydrolysates)? That could provide some indication as to whether the enhanced effects exerted by jellies+protein hydrolysate can be attributed to gelatin degradation.
Response: We agree with the reviewer and we have moved this text now to the discussion section and have consolidated the discussion as follows: “The food carrier used in this study (jelly candies) proved effective at maintaining the bioactivity of the Porphyridium sp. hydrolysate and actually increased antihypertensive activity by ~ 20% compared to the hydrolysate alone in relation to SBP and DBP. This may be due to release of gelatin/collagen peptides following digestion in the SHR that could further enhance antihypertensive activity. However, we did not include hydrolysed gelatin as a control. A study carried out recently by Cao and colleagues [51] identified that bovine gelatin-derived peptides (BGPs) reduced blood pressure, triglyceride levels, and the low-density lipoprotein cholesterol/high-density lipoprotein cholesterol ratio in SHRs through downregulated angiotensin converting enzyme (ACE-1), angiotensin II (Ang II), and Ang II type 1 receptor (AT1R) levels and upregulation of Ang II type 2 receptor (AT2R) levels.”
- DISCUSSION:
(a) Line 247: “A satisfactory protein extract yield (11.90%) resulted” – Why is 11.90% considered “satisfactory”? Did the authors come across any info in the literature that support this statement?
Response: We have revised this sentence and section as follows: “Protein was extracted from Porphyridium sp. biomass using a combination of Viscozyme and Alcalase. This combination of enzymes was used previously to extract proteins from seaweed biomass including Chondrus crispus and indeed microalgae such as Nannochloropsis gaditana [30, 31]. A yield of 11.90% protein (dry weight basis) resulted. Previous studies reported protein yields using Alcalase alone of 35.1 % [32]. The enzymes Alcalase, Thermolysin and Bromelain are the most commonly used enzymes to generate bioactive peptides from biomass including algae [32].” We also edited reference [32] to Safi et al., 2017
(b) Lines 248-249: “Thermolysin and Bromelain are the most commonly used enzymes to generate bioactive peptides from biomass including algae” – Can the authors clarify why they did not use these two enzymes instead since they have been used commonly, even on algae?
Response: Please see earlier revision. We included Alcalase in this text as Alcalase is also commonly used for hydrolysis and was previously applied to Nannochloropsis sp. as reported by Safi and colleagues 2017 (reference [32]).
(c) Lines 240-247: “Chosen peptides were also chosen based on the presence of the amino acids phenylalanine (F), Tyrosine (Y) and Leucine (L)…” – It is unclear how choosing such sequences contribute towards the aim of the study to search for anti-inflammatory/anti-COX/anti-hypertension/pain-reducing peptides. Please improve the rationale statement.
Response: We have revised this text now as follows on lines 261-265: “Chosen peptides were also selected based on the presence of the amino acids phenylalanine (F), Tyrosine (Y) and Leucine (L). The side chains of phenylalanine and leucine are long carbon chains or rings, which makes them bulky. Branched chain amino acids are essential for cellular signaling function and when found in peptides they may be ideal for disease prevention [37].” We also included the reference [37] as follows: Dullius, A., Fassina, P., Giroldi, M., Goettert, M. I., Volken de Souza, C. F. (2020). A biotechnological approach for the production of branched chain amino acid containing bioactive peptides to improve human health: A review. Food Research International, 131, https:doi.org/10.1016/j.foodres.2020.109002.
- M&M: Please include a short description/section to provide information about the statistical tests that were conducted, the software used, etc.
Response: 2.5 Statistical Analysis
All the experiments were performed in triplicates, and results are expressed as the mean ± standard deviation (SD) of the two replicates using EXCEL 2010. All measurements were carried out either in triplicate (n = 3) or (n=9). Statistical analysis was performed using Excel 2010. one-way ANOVA and a post-hoc Tukey’s HSD test was applied. In all cases, the criterion for statistical significance was p < 0.05.

Reviewer 2 Report
The article is well structured and written in a professional manner. The article describes the the generation and characterisation of a bioactive hy-23 drolysate generated using Viscozyme® and Alcalase from the red microalga Porphyridium sp. It is important aspect to find alternative treatment remedies to treat inflammation and pain without causing too much side effects to patients. However, many formatting mistakes are seen. It is suggested to go through the article and improve its formatting mistakes.
I recommend it for publication after minor revisions.
Author Response
Comments and Suggestions for Authors
The article is well structured and written in a professional manner. The article describes the the generation and characterisation of a bioactive hy-23 drolysate generated using Viscozyme® and Alcalase from the red microalga Porphyridium sp. It is important aspect to find alternative treatment remedies to treat inflammation and pain without causing too much side effects to patients. However, many formatting mistakes are seen. It is suggested to go through the article and improve its formatting mistakes.
I recommend it for publication after minor revisions.
Response: We thank the reviewer for their comments. We have revised the text in terms of formatting mistakes throughout and we have used spell check throughout. We deleted redundant information that was repetition from the results and discussion sections. We removed the conclusion and joined it with the discussion section.
